# Approximation-Aware Bayesian Optimization

**Natalie Maus**
University of Pennsylvania
nmaus@seas.upenn.edu

**Kyurae Kim**
University of Pennsylvania

**Geoff Pleiss**
University of British Columbia
Vector Institute

**David Eriksson**
Meta

**John P. Cunningham**
Columbia University

**Jacob R. Gardner**
University of Pennsylvania

## Abstract

High-dimensional Bayesian optimization (BO) tasks such as molecular design often require >10,000 function evaluations before obtaining meaningful results. While methods like sparse variational Gaussian processes (SVGPs) reduce computational requirements in these settings, the underlying approximations result in suboptimal data acquisitions that slow the progress of optimization. In this paper we modify SVGPs to better align with the goals of BO: targeting informed data acquisition rather than global posterior fidelity. Using the framework of utility-calibrated variational inference, we unify GP approximation and data acquisition into a joint optimization problem, thereby ensuring optimal decisions under a limited computational budget. Our approach can be used with any decision-theoretic acquisition function and is readily compatible with trust region methods like TuRBO. We derive efficient joint objectives for the expected improvement and knowledge gradient acquisition functions for standard and batch BO. Our approach outperforms standard SVGPs on high-dimensional benchmark tasks in control and molecular design.

## 1 Introduction

Bayesian optimization (BO; Frazier, 2018; Garnett, 2023; Jones et al., 1998; Mockus, 1982; Shahriari et al., 2015) casts optimization as a sequential decision-making problem. Many recent successes of BO have involved complex and high-dimensional problems. In contrast to "classic" low-dimensional BO problems—where expensive black-box function evaluations far exceeded computational costs—these modern problems necessitate tens of thousands of function evaluations, and it is often the complexity and dimensionality of the search space that makes optimization challenging, rather than a limited evaluation budget (Eriksson et al., 2019; Griffiths and Hernández-Lobato, 2020; Maus et al., 2022, 2023; Stanton et al., 2022). Because of these scenarios, BO is entering a regime where computational costs are becoming a primary bottleneck (Maddox et al., 2021; Maus et al., 2023; Moss et al., 2023; Vakili et al., 2021), as the Gaussian process (GP; Rasmussen and Williams, 2005) surrogate models that underpin most of Bayesian optimization scale cubically with the number of observations.

In this new regime, we require scalable GP approximations, an area that has made tremendous progress over the last decade. In particular, sparse variational Gaussian processes (SVGP; Hensman et al., 2013; Quiñonero-Candela and Rasmussen, 2005; Titsias, 2009) have seen an increase in use (Griffiths and Hernández-Lobato, 2020; Maddox et al., 2021; Maus et al., 2022, 2023; Stanton et al., 2022; Tripp et al., 2020; Vakili et al., 2021), but many challenges remain to effectively deploy SVGPs for large-budget BO. In particular, the standard SVGP training objective is not aligned with the goals of black-box optimization. SVGPs construct an inducing point approximation that maximizes the standard variational evidence lower bound (ELBO; Jordan et al., 1999), yielding a posterior approximation $q^*(f)$ that models all observed data (Matthews et al., 2016; Moss et al., 2023). However, the optimal posterior approximation $q^*$ is suboptimal for the decision-making tasks involved

38th Conference on Neural Information Processing Systems (NeurIPS 2024).

in BO (Lacoste–Julien et al., 2011). In BO, we do not care about posterior fidelity at the majority of prior observations; rather, we only care about the fidelity of downstream functions involving the posterior, such as the expected utility. To illustrate this point intuitively, consider using the common expected improvement (EI; Jones et al., 1998) acquisition function for selecting new observations. Maximizing the ELBO might result in a posterior approximation that maintains fidelity for training examples in regions of virtually zero EI, thus wasting "approximation budget."

To solve this problem, we focus on the deep connections between statistical decision theory (Robert, 2001; Wasserman, 2013, §12) and Bayesian optimization (Garnett, 2023, §6-7), where acquisition maximization can be viewed as maximizing posterior-expected utility. Following this perspective, we leverage the utility-calibrated approximate inference framework (Jaiswal et al., 2020, 2023; Lacoste–Julien et al., 2011), and solve the aforementioned problem through a variational bound (Blei et al., 2017; Jordan et al., 1999)–the (log) **expected utility lower bound (EULBO)**—a joint function of the decision (the BO query) and the posterior approximation (the SVGP). When optimized jointly, the EULBO automatically yields the approximately optimal decision through the minorize-maximize principle (Lange, 2016). The EULBO is reminiscent of the standard variational ELBO (Jordan et al., 1999), and can indeed be viewed as a standard ELBO for a generalized Bayesian inference problem (Bissiri et al., 2016; Knoblauch et al., 2022), where we seek to approximate the *utility-weighted* posterior. This work represents the first application of utility-calibrated approximate inference towards BO despite its inherent connection with utility maximization.

The benefits of our proposed approach are visualized in Fig. 1. Furthermore, it can be applied to acquisition function that admits a decision-theoretic interpretation, which includes the popular expected improvement (EI; Jones et al., 1998) and knowledge gradient (KG; Wu et al., 2017) acquisition functions, and is trivially compatible with local optimization techniques like TuRBO (Eriksson et al., 2019) for high-dimensional problems. We demonstrate that our joint SVGP/acquisition optimization approach yields significant improvements across numerous Bayesian optimization benchmarks. As an added benefit, our approach can simplify the implementation and reduce the computational burden of complex (decision-theoretic) acquisition functions like KG. We demonstrate a novel algorithm derived from our joint optimization approach for computing and optimizing the KG that expands recent work on one-shot KG (Balandat et al., 2020) and variational GP posterior refinement (Maddox et al., 2021).

Overall, our contributions are summarized as follows:

- We propose utility-calibrated variational inference of SVGPs in the context of large-budget BO.

- We study this framework in two special cases using the utility functions of two common acquisition functions: EI and KG. For each, we derive tractable EULBO expressions that can be optimized.

- For KG, we demonstrate that the computation of the EULBO takes only negligible additional work over computing the standard ELBO by leveraging an online variational update. Thus, as a byproduct of optimizing the EULBO, optimizing KG becomes comparable to the cost of the EI.

- We extend this framework to be capable of running in batch mode, by introducing q-EULBO analogs of q-KG and q-EI as commonly used in practice (Wilson et al., 2018).

- We demonstrate the effectiveness of our proposed method against standard SVGPs trained with ELBO maximization on high-dimensional benchmark tasks in control and molecular design, where the dimensionality and evaluation budget go up to 256 and 80k, respectively.

## 2 Background

**Noisy Black-Box Optimization.** Noisy black-box optimization refers to problems of the form: $\text{maximize}_{\boldsymbol{x} \in \mathcal{X}} \, F(\boldsymbol{x})$, where $\mathcal{X} \subset \mathbb{R}^d$ is some compact domain, $F : \mathcal{X} \to \mathcal{Y}$ is some objective function, and we assume that only zeroth-order information of $F$ is available. More formally, for some $i \in \mathbb{N}_{>0}$, we assume that observations of the objective function $(\boldsymbol{x}_i, y_i = \widehat{F}(\boldsymbol{x}_i))$ have been corrupted by independently and identically distributed (i.i.d.) Gaussian noise $\widehat{F}(\boldsymbol{x}_i) \triangleq F(\boldsymbol{x}_i) + \epsilon$, where $\epsilon \sim \mathcal{N}(0, \sigma_\text{n}^2)$. The noise variance $\sigma_\text{n}^2$ is also unknown.

**Bayesian optimization.** Bayesian Optimization (BO) is and iterative approach to noisy black-box optimization that iterates the following steps: ❶ At each step $t \geq 0$, we use a set of observations $\mathcal{D}_t = \{(\boldsymbol{x}_i, y_i = \widehat{F}(\boldsymbol{x}_i))\}_{i=1}^{n_t}$ of $\widehat{F}$ to fit a surrogate supervised model $f \in \mathcal{F}$. Typically, $\mathcal{F}$ is taken to be the sample space of a Gaussian process such that the function-valued posterior distribution

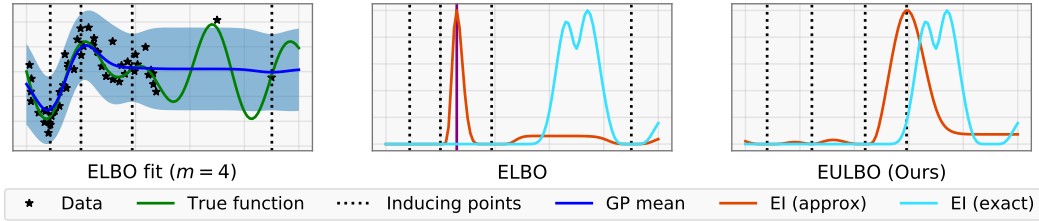

Figure 1: **(Left.)** Fitting an SVGP model with only $m = 4$ inducing points sacrifices modeling areas of high EI (few data points at right) because the ELBO focuses only on global data approximation (left data) and is ignorant of the downstream decision making task. **(Middle.)** Because of this, (normalized) EI with the SVGP model peaks in an incorrect location relative to the exact posterior. **(Right.)** Updating the GP fit and selecting a candidate jointly using the EULBO (our method) results in candidate selection much closer to the exact model.

$\pi(f \mid \mathcal{D})$ forms a distribution over surrogate models at step $t$. ❷ The posterior is then used to form a decision problem where we choose which point we should evaluate next, $\boldsymbol{x}_{t+1} = \delta_\alpha(\mathcal{D}_t)$, by maximizing an acquisition function $\alpha : \mathcal{X} \to \mathbb{R}$ as

$$\delta_\alpha(\mathcal{D}_t) \triangleq \arg\max_{\boldsymbol{x} \in \mathcal{X}} \ \alpha(\boldsymbol{x}; \mathcal{D}_t). \tag{1}$$

❸ After selecting $\boldsymbol{x}_{t+1}$, $\widehat{F}$ is evaluated to obtain the new datapoint $(\boldsymbol{x}_{t+1}, y_{t+1} = \widehat{F}(\boldsymbol{x}_{t+1}))$. This is then added to the dataset, forming $\mathcal{D}_{t+1} = \mathcal{D}_t \cup (\boldsymbol{x}_{t+1}, y_{t+1})$ to be used in the next iteration.

**Utility-Based Acquisition Functions.** Many commonly used acquisition functions, including EI and KG, can be expressed as posterior-expected utility functions

$$\alpha(\boldsymbol{x}; \mathcal{D}) \triangleq \int u(\boldsymbol{x}, f; \mathcal{D}) \pi(f \mid \mathcal{D}) \, \mathrm{d}f, \tag{2}$$

where $u(\boldsymbol{x}, f; \mathcal{D}) : \mathcal{X} \times \mathcal{F} \to \mathbb{R}$ is some utility function associated with $\alpha$ (Garnett, 2023, §6-7). In statistical decision theory, posterior-expected utility maximization policies such as $\delta_\alpha$ are known as *Bayes policies*. These are important because, for a given utility function, they attain certain notions of statistical optimality such as Bayes optimality and admissibility (Robert, 2001, §2.4; Wasserman, 2013, §12). However, this only holds true if we can exactly compute Eq. (2) over the posterior. Once approximate inference is involved, making optimal Bayes decisions becomes challenging.

**Sparse Variational Gaussian Processes.** While the $\mathcal{O}(n^3)$ complexity of exact Gaussian process model selection and inference is not necessarily a roadblock in the traditional regression setting with 10,000-50,000 training examples, BO amplifies the scalability challenge by requiring us to sequentially train or update *many* large scale GPs as we iteratively acquire more data.

To address this, sparse variational GPs (SVGP; Hensman et al., 2013; Titsias, 2009) have become commonly used in high-throughput Bayesian optimization. SVGPs modify the original GP prior from $p(f)$ to $p(f \mid \boldsymbol{u})p(\boldsymbol{u})$, where we assume the latent function $f$ is "induced" by a finite set of *inducing values* $\boldsymbol{u} = (u_1, \dots, u_m) \in \mathbb{R}^m$ located at *inducing points* $\boldsymbol{z}_i \in \mathcal{X}$ for $i = 1, \dots, m$. Inference is done through variational inference (Blei et al., 2017; Jordan et al., 1999), where the posterior of the inducing points is approximated using $q_\lambda(\boldsymbol{u}) = \mathcal{N}(\boldsymbol{u}; \lambda = (\boldsymbol{m}, \boldsymbol{S}))$ and that of the latent functions with $q(f \mid \boldsymbol{u}) = p(f \mid \boldsymbol{u})$. Here, the variational parameters $\boldsymbol{m}$ and $\boldsymbol{S}$ are defined as the learned mean and covariance of the variational distribution $q_\lambda(\boldsymbol{u})$. It is standard practice to define $\lambda = (\boldsymbol{m}, \boldsymbol{S})$ so that $\lambda$ can be used as shorthand to represent all of the trainable variational parameters. As is typical in the BO literature, we use the subscript $\lambda \in \Lambda$ to denote that the distribution denoted as $q$ contains trainable parameters in $\lambda$.

For a positive definite kernel function $k : \mathcal{X} \times \mathcal{X} \to \mathbb{R}_{>0}$, the resulting ELBO objective, which can be computed in a closed form (Hensman et al., 2013), is then

$$\mathcal{L}_{\mathrm{ELBO}}(\lambda; \mathcal{D}_t) \triangleq \mathbb{E}_{q_\lambda(f)}\left[\sum_{i=1}^{n_t} \log \ell(y_i \mid f(\boldsymbol{x}_i))\right] - \mathrm{D}_{\mathrm{KL}}(q_\lambda(\boldsymbol{u}), p(\boldsymbol{u})), \tag{3}$$

where $\ell(y_i \mid f(\boldsymbol{x}_i)) = \mathcal{N}(y_i \mid f(\boldsymbol{x}_i), \sigma_\varepsilon)$ is a Gaussian likelihood. The marginal variational approximation can be computed as

$$q_\lambda(f) = \int q_\lambda(f, \boldsymbol{u}) \, \mathrm{d}\boldsymbol{u} = \int p(f \mid \boldsymbol{u}) \, q_\lambda(\boldsymbol{u}) \, \mathrm{d}\boldsymbol{u}$$

such that the point-wise function evaluation on some $\boldsymbol{x} \in \mathcal{X}$ is

$$q_\lambda(f(\boldsymbol{x})) = \mathcal{N}\left(f(\boldsymbol{x}); \quad \mu_f(\boldsymbol{x}) \triangleq \boldsymbol{K}_{xZ}\boldsymbol{K}_{ZZ}^{-1}\boldsymbol{m}, \quad \sigma_f^2(\boldsymbol{x}) \triangleq \widetilde{k}_{xx} + \boldsymbol{k}_{xZ}^\top \boldsymbol{K}_{ZZ}^{-1}\boldsymbol{S}\boldsymbol{K}_{ZZ}^{-1}\boldsymbol{k}_{Zx}\right), \quad (4)$$

with $\widetilde{k}_{xx} \triangleq k(\boldsymbol{x}, \boldsymbol{x}) - \boldsymbol{k}_{xZ}\boldsymbol{K}_{ZZ}^{-1}\boldsymbol{k}_{Zx}^\top$, the vector $\boldsymbol{k}_{Zx} \in \mathbb{R}^m$ is formed as $[\boldsymbol{k}_{Zx}]_i = k(\boldsymbol{z}_i, \boldsymbol{x})$, and the matrix $\boldsymbol{K}_{ZZ} \in \mathbb{R}^{m\times m}$ is formed as $[\boldsymbol{K}_{ZZ}]_{ij} = k(\boldsymbol{z}_i, \boldsymbol{z}_j)$. Additionally, the GP likelihood and kernel contain hyperparameters, which we denote as $\theta \in \Theta$, and we collectively denote the set of inducing point locations as $\boldsymbol{Z} = (\boldsymbol{z}_1, \dots, \boldsymbol{z}_m) \in \mathcal{X}^m$. We therefore denote the ELBO as $\mathcal{L}_{\mathrm{ELBO}}(\lambda, \boldsymbol{Z}, \theta; \mathcal{D}_t)$.

## 3 Approximation-Aware Bayesian Optimization

When SVGPs are used in conjunction with BO (Maddox et al., 2021; Moss et al., 2023) at iteration $t \geq 0$, acquisition functions of the form of Eq. (2) are naïvely approximated as

$$\alpha(\boldsymbol{x}; \mathcal{D}) \approx \int u(\boldsymbol{x}, f; \mathcal{D}_t)\, q_\lambda(f)\, \mathrm{d}f,$$

where $q_\lambda(f)$ is the approximate SVGP posterior given by Eq. (4). The acquisition policy implied by this approximation contains two separate optimization problems:

$$\boldsymbol{x}_{t+1} = \arg\max_{\boldsymbol{x} \in \mathcal{X}} \int u(\boldsymbol{x}, f; \mathcal{D}_t)\, q_{\lambda^*_{\mathrm{ELBO}}}(f)\, \mathrm{d}f \quad \text{and} \quad \lambda^*_{\mathrm{ELBO}} = \arg\max_{\lambda \in \Lambda} \mathcal{L}_{\mathrm{ELBO}}(\lambda; \mathcal{D}_t). \quad (5)$$

Treating these optimization problems separately creates an artificial bottleneck that results in suboptimal data acquisition decisions. Intuitively, $\lambda^*_{\mathrm{ELBO}}$ is chosen to faithfully model all observed data (Matthews et al., 2016; Moss et al., 2023), without regard for how the resulting model performs at selecting the next function evaluation in the BO loop. For an illustration of this, see Figure 1. Instead, we propose a modification to SVGPs that couples the posterior approximation and data acquisition through a joint problem of the form:

$$(\boldsymbol{x}_{t+1}, \lambda^*) = \arg\max_{\lambda \in \Lambda, \boldsymbol{x} \in \mathcal{X}} \mathcal{L}_{\mathrm{EULBO}}(\lambda, \boldsymbol{x}; \mathcal{D}_t). \quad (6)$$

This results in $\boldsymbol{x}_{t+1}$ directly approximating a solution to Eq. (2), where the **expected utility lower-bound** (EULBO) is an ELBO-like objective function derived below.

### 3.1 Expected Utility Lower-Bound

Consider an acquisition function of the form of Eq. (2), where the utility $u : \mathcal{X} \times \mathcal{F} \to \mathbb{R}_{>0}$ is strictly positive. We can derive a similar variational formulation of the acquisition function maximization problem following Lacoste–Julien et al. (2011). That is, given any distribution $q_\lambda$ indexed by $\lambda \in \Lambda$ and considering the SVGP prior augmentation $p(f) \to p(f \mid \boldsymbol{u})p(\boldsymbol{u})$, the acquisition function can be lower-bounded through Jensen's inequality as

$$
\begin{aligned}
\log \alpha(\boldsymbol{x}; \mathcal{D}_t) &= \log \int u(\boldsymbol{x}, f; \mathcal{D}_t)\, \pi(f \mid \mathcal{D}_t)\, \mathrm{d}f \\
&= \log \int u(\boldsymbol{x}, f; \mathcal{D}_t)\, \pi(f, \boldsymbol{u} \mid \mathcal{D}_t)\, \frac{q_\lambda(f, \boldsymbol{u})}{q_\lambda(f, \boldsymbol{u})}\, \mathrm{d}f\, \mathrm{d}\boldsymbol{u} \\
&= \log \int u(\boldsymbol{x}, f; \mathcal{D}_t)\, \ell(\mathcal{D}_t \mid f)\, p(f \mid \boldsymbol{u})\, p(\boldsymbol{u})\, \frac{q_\lambda(\boldsymbol{u})\, p(f \mid \boldsymbol{u})}{q_\lambda(\boldsymbol{u})\, p(f \mid \boldsymbol{u})}\, \mathrm{d}f\, \mathrm{d}\boldsymbol{u} - \log Z \\
&\geq \int \log\left(\frac{u(\boldsymbol{x}, f; \mathcal{D}_t)\, \ell(\mathcal{D}_t \mid f)\, p(\boldsymbol{u})}{q_\lambda(\boldsymbol{u})}\right) p(f \mid \boldsymbol{u})\, q_\lambda(\boldsymbol{u})\, \mathrm{d}f\, \mathrm{d}\boldsymbol{u} - \log Z, \quad (7)
\end{aligned}
$$

where $Z$ is a normalizing constant. A restriction on $u$ comes from the inequality in Eq. (7), where the utility needs to be strictly positive. This means that non-strictly positive utilities need to be modified to be incorporated into this framework. (See the examples by Kuśmierczyk et al., 2019.) Also, notice that the derivation is reminiscent of expectation-maximization (Dempster et al., 1977) and variational lower bounds (Jordan et al., 1999). That is, through the minorize-maximize principle (Lange, 2016), maximizing the lower bound with respect to $\boldsymbol{x}$ and $\lambda$ approximately solves the original problem of maximizing the posterior-expected utility.

**Expected Utility Lower-Bound.** Up to a constant and rearranging terms, maximizing Eq. (7) is equivalent to maximizing

$$\mathcal{L}_{\text{EULBO}}(\boldsymbol{\lambda}, \boldsymbol{x}; \mathcal{D}_t) \triangleq \mathbb{E}_{p(f|\boldsymbol{u})q_{\boldsymbol{\lambda}}(\boldsymbol{u})}\left[\log \ell(\mathcal{D}_t \mid f) + \log p(\boldsymbol{u}) - \log q_{\boldsymbol{\lambda}}(\boldsymbol{u}) + \log u(\boldsymbol{x}, f; \mathcal{D}_t)\right]$$

$$= \mathbb{E}_{q_{\boldsymbol{\lambda}}(f)}\left[\sum_{i=1}^{n_t} \log \ell(y_i \mid f)\right] - \mathrm{D}_{\mathrm{KL}}(q_{\boldsymbol{\lambda}}(\boldsymbol{u}), p(\boldsymbol{u})) + \mathbb{E}_{q_{\boldsymbol{\lambda}}(f)} \log u(\boldsymbol{x}, f; \mathcal{D}_t)$$

$$= \mathcal{L}_{\text{ELBO}}(\boldsymbol{\lambda}; \mathcal{D}_t) + \mathbb{E}_{q_{\boldsymbol{\lambda}}(f)} \log u(\boldsymbol{x}, f; \mathcal{D}_t), \qquad (8)$$

which is the joint objective function alluded to in Eq. (6). We maximize EULBO to obtain $(\boldsymbol{x}_{t+1}, \boldsymbol{\lambda}^*) = \arg\max_{\boldsymbol{x}\in\mathcal{X}, \boldsymbol{\lambda}\in\Lambda} \mathcal{L}_{\text{EULBO}}(\boldsymbol{x}, \boldsymbol{\lambda})$, where $\boldsymbol{x}_{t+1}$ corresponds our next BO "query".

From Eq. (8), the connection between the EULBO and ELBO is obvious: the EULBO is now "nudging" the ELBO solution toward high utility regions. An alternative perspective is that we are approximating a *generalized posterior* weighted by the utility (Table. 1 by Knoblauch et al., 2022; Bissiri et al., 2016). Furthermore, Jaiswal et al. (2020, 2023) prove that the resulting actions satisfy consistency guarantees under assumptions typical in such results for variational inference (Wang and Blei, 2019).

**Hyperparameters and Inducing Point Locations.** For the hyperparameters $\boldsymbol{\theta}$ and inducing point locations $\boldsymbol{Z}$, we use the marginal likelihood to perform model selection, which is common practice in BO (Shahriari et al., 2015, §V.A). (Optimizing over $\boldsymbol{Z}$ was popularized by Snelson and Ghahramani, 2005.) Following suit, we also optimize the EULBO as a function of $\boldsymbol{\theta}$ and $\boldsymbol{Z}$ as

$$\underset{\boldsymbol{\lambda}, \boldsymbol{x}, \boldsymbol{\theta}, \boldsymbol{Z}}{\text{maximize}} \left\{ \mathcal{L}_{\text{EULBO}}(\boldsymbol{\lambda}, \boldsymbol{x}, \boldsymbol{\theta}, \boldsymbol{Z}; \mathcal{D}_t) \triangleq \mathcal{L}_{\text{ELBO}}(\boldsymbol{\lambda}, \boldsymbol{Z}, \boldsymbol{\theta}; \mathcal{D}_t) + \mathbb{E}_{q_{\boldsymbol{\lambda}}(f)} \log u(\boldsymbol{x}, f; \mathcal{D}_t) \right\}.$$

We emphasize here that the SVGP-associated parameters $\boldsymbol{\lambda}, \boldsymbol{\theta}, \boldsymbol{Z}$ have gradients that are determined by *both* terms above. Thus, the expected log-utility term $\mathbb{E}_{f\sim q_{\boldsymbol{\lambda}}(f)} \log u(\boldsymbol{x}, f; \mathcal{D}_t)$ simultaneously results in acquisition of $\boldsymbol{x}_{t+1}$ and directly influences the underlying SVGP regression model.

## 3.2 EULBO for Expected Improvement (EI)

The EI acquisition function can be expressed as a posterior-expected utility, where the underlying "improvement" utility function is given by the difference between the objective value of the query, $f(\boldsymbol{x})$, and the current best objective value $y_t^* = \max_{i=1,\dots,t}\{y_i \mid y_i \in \mathcal{D}_t\}$:

$$u_{\text{EI}}(\boldsymbol{x}, f; \mathcal{D}_t) \triangleq \text{ReLU}\left(f(\boldsymbol{x}) - y_t^*\right), \qquad (\text{EI; Jones et al., 1998}) \qquad (9)$$

where $\text{ReLU}(x) \triangleq \max(x, 0)$. Unfortunately, this utility is not strictly positive whenever $f(\boldsymbol{x}) \le y^*$. Thus, we cannot immediately plug $u_{\text{EI}}$ into the EULBO. While it is possible to add a small positive constant to $u_{\text{EI}}$ and make it strictly positive as done by Kuśmierczyk et al. (2019), this results in a looser Jensen gap in Eq. (7), which could be detrimental. This also introduces the need for tuning the constant, which is not straightforward. Instead, we define the following "soft" EI utility:

$$u_{\text{SEI}}(\boldsymbol{x}, f; \mathcal{D}_t) \triangleq \text{softplus}\left(f(\boldsymbol{x}) - y_t^*\right),$$

where the ReLU in Eq. (9) is replaced with $\text{softplus}(x) \triangleq \log(1 + \exp(x))$. $\text{softplus}(x)$ converges to the ReLU in both extremes of $x \to \pm\infty$. Thus, $u_{\text{SEI}}$ will behave closely to $u_{\text{EI}}$, while being slightly more explorative due to positivity.

Computing the EULBO and its derivatives now requires the computation of $\mathbb{E}_{f\sim q_{\boldsymbol{\lambda}}(f)} \log u_{\text{SEI}}(\boldsymbol{x}, f; \mathcal{D}_t)$, which, unlike EI, does not have a closed-form. However, since the utility function only depends on the function values of $f$, the expectation can be efficiently computed to high precision through one-dimensional Gauss-Hermite quadrature. Crucially, the expensive $K_{zz}^{-1}m$ and $K_{zz}^{-1}SK_{zz}^{-1}$ solves that dominate both the asymptotic and practical running time of both the ELBO and the EULBO are fixed across the log utility evaluations needed by quadrature. Because quadrature only depends on these precomputed moments, the additional work necessary due to lacking a closed form solution is negligible: Gauss-Hermite quadrature converges extremely quickly in the number of quadrature sites, and only requires on the order of 10 or so of these post-solve evaluations to achieve near machine precision.

## 3.3 EULBO for Knowledge Gradient (KG)

Although non-trivial, the KG acquisition is also a posterior-expected utility, where the underlying utility function is given by the maximum predictive mean value anywhere in the input domain *after*

conditioning on a new observation $(\boldsymbol{x}, y) \in \mathcal{X} \times \mathcal{Y}$:

$$u_{\text{KG}}(\boldsymbol{x}, y; \mathcal{D}_t) \triangleq \max_{\boldsymbol{x}' \in \mathcal{X}} \mathbb{E}\left[f(\boldsymbol{x}') \mid \mathcal{D}_t \cup \{(\boldsymbol{x}, y)\}\right]. \qquad (\text{KG; Frazier, 2009; Garnett, 2023})$$

Note that the utility function as defined above is not non-negative: the maximum predictive mean of a Gaussian process can be negative. For this reason, the utility function is commonly (and originally, *e.g.* Frazier, 2009, Eq. 4.11) written in the literature as the *difference* between the new maximum mean after conditioning on $(\boldsymbol{x}, y)$ and the maximum mean beforehand:

$$u_{\text{KG}}(\boldsymbol{x}, y; \mathcal{D}_t) \triangleq \max_{\boldsymbol{x}' \in \mathcal{X}} \mathbb{E}\left[f(\boldsymbol{x}') \mid \mathcal{D}_t \cup \{(\boldsymbol{x}, y)\}\right] - \mu_t^+,$$

where $\mu_t^+ \triangleq \max_{\boldsymbol{x}'' \in \mathcal{X}} E\left[f(\boldsymbol{x}'') \mid \mathcal{D}_t\right]$. Note that $\mu_t^+$ plays the role of a simple constant as it depends on neither $\boldsymbol{x}$ nor $y$. Similarly to the EI acquisition, this utility is still not strictly positive, and we thus define its "softplus-ed" variant:

$$u_{\text{SKG}}(\boldsymbol{x}, y; \mathcal{D}_t) \triangleq \text{softplus}\left(u_{\text{KG}}(\boldsymbol{x}, y; \mathcal{D}_t) - c^+\right).$$

Here, $c^+$ acts as $\mu_t^+$ by making $u_{\text{KG}}$ positive as often as possible. This is particularly important when the GP predictive mean is negative as a consequence of the objective values being negative. One natural choice of constant is using $\mu_t^+$; however, we find that simply choosing $c^+ = y_t^+$ works well and is more computationally efficient. Here, $y_t^+$ is the highest value of $y_t$ (the highest objective value observed so far).

**One-Shot KG EULBO.** The EULBO using $u_{\text{SKG}}$ results in an expensive nested optimization problem. To address this, we use an approach similar to the one-shot knowledge gradient method of Balandat et al. (2020). For clarity, we will define the reparameterization function

$$y_\lambda(\boldsymbol{x}; \epsilon_i) \triangleq \mu_{q_\lambda}(\boldsymbol{x}) + \sigma_{q_\lambda}(\boldsymbol{x})\epsilon_i,$$

where, for an i.i.d. sample $\epsilon_i \sim \mathcal{N}(0, 1)$, computing $y_i = y_\lambda(\boldsymbol{x}, \epsilon_i)$ is equivalent to sampling $y_i \sim \mathcal{N}\left(\mu_{q_\lambda}(\boldsymbol{x}), \sigma_{q_\lambda}(\boldsymbol{x})\right)$. This enables the use of the reparameterization gradient estimator (Kingma and Welling, 2014; Rezende et al., 2014; Titsias and Lázaro-Gredilla, 2014). Now, notice that the KG acquisition function can be approximated through Monte Carlo as

$$\alpha_{\text{KG}}(\boldsymbol{x}; \mathcal{D}) \approx \frac{1}{S}\sum_{i=1}^{S} u_{\text{KG}}(\boldsymbol{x}, y_\lambda(\boldsymbol{x}; \epsilon_i); \mathcal{D}_t) = \frac{1}{S}\sum_{i=1}^{S} \max_{\boldsymbol{x}'} \mathbb{E}\left[f(\boldsymbol{x}') \mid \mathcal{D}_t \cup \{\boldsymbol{x}, y_\lambda(\boldsymbol{x}; \epsilon_i)\}\right],$$

where, for $i = 1, \dots, S$, $\epsilon_i \sim \mathcal{N}(0, 1)$ are i.i.d. The one-shot KG approach absorbs the nested optimization over $\boldsymbol{x}'$ into a simultaneous joint optimization over $\boldsymbol{x}$ and a mean maximizer for each of the S samples, $\boldsymbol{x}'_1, \dots, \boldsymbol{x}'_S$ such that $\max_{\boldsymbol{x}} \alpha_{\text{KG}}(\boldsymbol{x}; \mathcal{D}_t) \approx \max_{\boldsymbol{x}, \boldsymbol{x}'_1, \dots, \boldsymbol{x}'_S} \alpha_{\text{1-KG}}(\boldsymbol{x}; \mathcal{D})$, where

$$\alpha_{\text{1-KG}}(\boldsymbol{x}; \mathcal{D}_t) \triangleq \frac{1}{S}\sum_{i=1}^{S} u_{\text{1-KG}}(\boldsymbol{x}, \boldsymbol{x}'_i, y_\lambda(\boldsymbol{x}; \epsilon_i); \mathcal{D}_t) = \frac{1}{S}\sum_{i=1}^{S} \mathbb{E}\left[f(\boldsymbol{x}'_i) \mid \mathcal{D}_t \cup \{\boldsymbol{x}, y_\lambda(\boldsymbol{x}; \epsilon_i)\}\right],$$

Evidently, there is no longer an inner optimization problem over $\boldsymbol{x}'$. To estimate the $i$th term of this sum, we draw a sample of the objective value of $\boldsymbol{x}$, $y_\lambda(\boldsymbol{x}; \epsilon_i)$, and condition the model on this sample. We then compute the new posterior predictive mean at $\boldsymbol{x}'_i$. After summing, we compute gradients with respect to both the candidate $\boldsymbol{x}$ and the mean maximizers $\boldsymbol{x}'_1, \dots, \boldsymbol{x}'_S$. Again, we use the "soft" version of one-shot KG in our EULBO optimization problem:

$$u_{\text{1-SKG}}(\boldsymbol{x}, \boldsymbol{x}', y; \mathcal{D}_t) = \text{softplus}\left(\mathbb{E}\left[f(\boldsymbol{x}') \mid \mathcal{D}_t \cup \{(\boldsymbol{x}, y)\}\right] - c^+\right),$$

where this utility function is crucially a function of both $\boldsymbol{x}$ and a free parameter $\boldsymbol{x}'$. As with $\alpha_{\text{1-KG}}$, maximizing the EULBO can be set up as a joint optimization problem:

$$\underset{\boldsymbol{x}, \boldsymbol{x}'_1, \dots, \boldsymbol{x}'_S, \lambda, \boldsymbol{Z}, \theta}{\text{maximize}} \quad \mathcal{L}_{\text{ELBO}}(\lambda, \boldsymbol{Z}, \theta) + \frac{1}{S}\sum_{i=1}^{S} \log u_{\text{1-SKG}}(\boldsymbol{x}, \boldsymbol{x}'_i, y_\lambda(\boldsymbol{x}; \epsilon_i); \mathcal{D}_t) \qquad (10)$$

**Efficient KG-EULBO Computation.** The computation time of the non-ELBO term in Eq. (10) is dominated by having to compute $\mathbb{E}\left[f(\boldsymbol{x}'_i) \mid \mathcal{D}_t \cup \{(\boldsymbol{x}, y_\lambda(\boldsymbol{x}; \epsilon_i))\}\right]$ $S$-times. Notice that we only need to compute an updated posterior predictive mean, and can ignore predictive variances. For this, we can leverage the online updating strategy of Maddox et al. (2021). In particular, the predictive mean can be updated in $\mathcal{O}(m^2)$ time using a simple Cholesky update. The additional $\mathcal{O}(Sm^2)$ cost of computing the EULBO is therefore amortized by the original $\mathcal{O}(m^3)$ cost of computing the ELBO.

### 3.4 Extension to `q-EULBO` for Batch Bayesian Optimization

The `EULBO` can be extended to support batch Bayesian optimization by using the Monte Carlo batch mode analogs of utility functions as discussed *e.g.* by Balandat et al. (2020); Wilson et al. (2018). Given a set of candidates $\boldsymbol{X} = (\boldsymbol{x}_1, ..., \boldsymbol{x}_q) \in \mathcal{X}^q$, the q-EI utility function is given by:

$$u_{q\text{-EI}}(\boldsymbol{X}, \boldsymbol{f}; \mathcal{D}_t) \triangleq \max_{j=1...q} \text{ReLU}\left(f(\boldsymbol{x}_j) - y_t^*\right) \quad \text{(q-EI; Balandat et al., 2020; Wilson et al., 2018)}$$

This utility can again be softened as:

$$u_{q\text{-SEI}}(\boldsymbol{X}, \boldsymbol{f}; \mathcal{D}_t) \triangleq \max_{j=1...q} \text{softplus}\left(f(\boldsymbol{x}_j) - y_t^*\right)$$

Because this is now a $q$-dimensional integral, Gauss-Hermite quadrature is no longer applicable. However, we can apply Monte Carlo as

$$\mathbb{E}_{q_\lambda(f)} \log u_{q\text{-SEI}}(\boldsymbol{X}, \boldsymbol{f}; \mathcal{D}_t) \approx \frac{1}{S} \sum_{i=1}^{S} \max_{j=1...q} \text{softplus}\left(y_\lambda(\boldsymbol{x}; \epsilon_i) - y_t^*\right).$$

As done in the BoTorch software package (Balandat et al., 2020), we observe that fixing the set of base samples $\epsilon_1, ..., \epsilon_S$ during each BO iteration results in better optimization performance at the cost of negligible `q-EULBO` bias. Now, optimizing the `q-EULBO` is done over the full set of $q$ candidates $(\boldsymbol{x}_1, ..., \boldsymbol{x}_q)$ jointly, as well as the GP hyperparameters, inducing points, and variational parameters.

**Knowledge Gradient.** The KG version of the `EULBO` can be similarly extended. The expected log utility term in the maximization problem Eq. (10) becomes:

$$\underset{\boldsymbol{x}_1,...,\boldsymbol{x}_q, \boldsymbol{x}_1',...,\boldsymbol{x}_S', \lambda, \boldsymbol{Z}, \theta}{\text{maximize}} \mathcal{L}_{\text{ELBO}}(\lambda, \boldsymbol{Z}, \theta) + \frac{1}{S} \sum_{i=1}^{S} \max_{j=1..q} \log u_{1\text{-SKG}}(\boldsymbol{x}_j, \boldsymbol{x}_i', y_\lambda(\boldsymbol{x}; \epsilon_i); \mathcal{D}_t),$$

resulting in a similar analog to q-KG as described by Balandat et al. (2020).

### 3.5 Optimizing the `EULBO`

Optimizing the ELBO for SVGPs is known to be challenging (Galy-Fajou and Opper, 2021; Terenin et al., 2024) as the optimization landscape for the inducing points is non-convex, multi-modal, and non-smooth. Naturally, these are also challenges for `EULBO`; we found that care must be taken when implementing and initializing the `EULBO` maximization problem. In this subsection, we outline some key ideas, while a detailed description with pseudocode is presented in Appendix A.

**Initialization and Warm-Starting.** We warm-start the `EULBO` maximization procedure by solving the conventional two-step scheme in Eq. (5): At each BO iteration, we obtain the "warm" initial values for $(\lambda, \boldsymbol{Z}, \theta)$ by optimizing the standard ELBO. Then, we use this to maximize the conventional acquisition function corresponding to the chosen utility function $u$ (the expectation of $u$ over $q_\lambda(f)$), which provides the warm-start initialization for $\boldsymbol{x}$.

**Alternating Maximization Scheme.** To optimize $\mathcal{L}_{\text{EULBO}}(\boldsymbol{x}, \lambda, \boldsymbol{Z}, \theta)$, we alternate between optimizing over the query $\boldsymbol{x}$ and the SVGP parameters $\lambda, \boldsymbol{Z}, \theta$. We find this block-coordinate descent scheme to be more stable and robust than jointly updating all parameters, though the reason why this is more stable than jointly optimizing all parameters requires further investigation.

## 4 Experiments

We evaluate `EULBO`-based SVGPs on a number of benchmark BO tasks, described in detail in Section 4.1. These tasks include standard low-dimensional BO problems, e.g., the 6D Hartmann function, as well as 7 high-dimensional and high-throughput optimization tasks.

**Baselines.** We compare `EULBO` to several baselines with the main goal of achieving a high reward using as few function evaluations as possible. Our primary point of comparison is ELBO-based SVGPs. We consider two approaches for inducing point locations: 1. optimizing inducing point locations via the ELBO (denoted as **ELBO**), 2. placing the inducing points using the strategy proposed by Moss et al. (2023) at each stage of ELBO optimization (denoted as **Moss et al.**). The latter offers improved BO performance over standard ELBO-SVGP in BO settings, yet—unlike our method—it exclusively

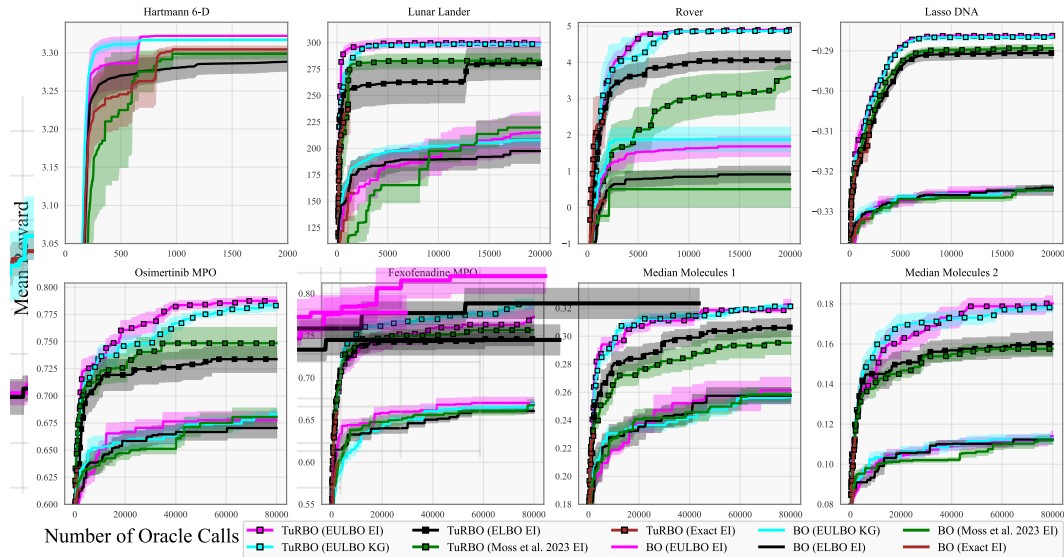

Figure 2: **Optimization results on the 8 considered tasks.** We compare all methods for both standard BO and TuRBO-based BO (on all tasks except Hartmann). Each line/shaded region represents the mean/standard error over 20 runs See subsection B.1 for additional molecule results.

targets inducing point placement and does not affect variational parameters or hyperparameters of the model. In addition, we compare to BO using exact GPs using 2, 000 function evaluations as the use of exact GP is intractable beyond this point due to the need to *repeatedly* fit models.

**Acquisition Functions and BO algorithms.** For EULBO, we test the versions based on both the Expected Improvement (EI) and Knowledge Gradient (KG) acquisition functions as well as the batch variant. We test the baseline methods using EI only. On high-dimensional tasks (tasks with dimensionality above 10), we run EULBO and baseline methods with standard BO and with trust region Bayesian optimization (TuRBO) (Eriksson et al., 2019). For the largest tasks (Lasso, Molecules) we use acquisition batch size of 20 ($q = 20$), and batch size 1 ($q = 1$) for all others.

**Implementation Details and Hyperparameters.** Code to reproduce all results in the paper is available at https://github.com/nataliemaus/aabo. We implement EULBO and baseline methods using the GPyTorch (Gardner et al., 2018) and BoTorch (Balandat et al., 2020) packages. For all methods, we initialize using a set of 100 data points sampled uniformly at random in the search space. We use the same trust region hyperparameters as in (Eriksson et al., 2019). In Appendix B.1, we also evaluate an additional initialization strategy for the molecular design tasks. This alternative initialization matches prior work in using 10, 000 molecules from the GuacaMol dataset Brown et al. (2019) rather than the details we used above for consistency across tasks, but does achieve higher overall performance.

## 4.1 Tasks

**Hartmann 6D.** The widely used Hartmann benchmark function (Surjanovic and Bingham, 2013).

**Lunar Lander.** The goal of this task is to find an optimal 12-dimensional control policy that allows an autonomous lunar lander to consistently land without crashing. The final objective value we optimize is the reward obtained by the policy averaged over a set of 50 random landing terrains. For this task, we use the same controller setup used by Eriksson et al. (2019).

**Rover.** The rover trajectory optimization task introduced by Wang et al. (2018) consists of finding a 60-dimensional policy that allows a rover to move along some trajectory while avoiding a set of obstacles. We use the same obstacle set up as in Maus et al. (2023).

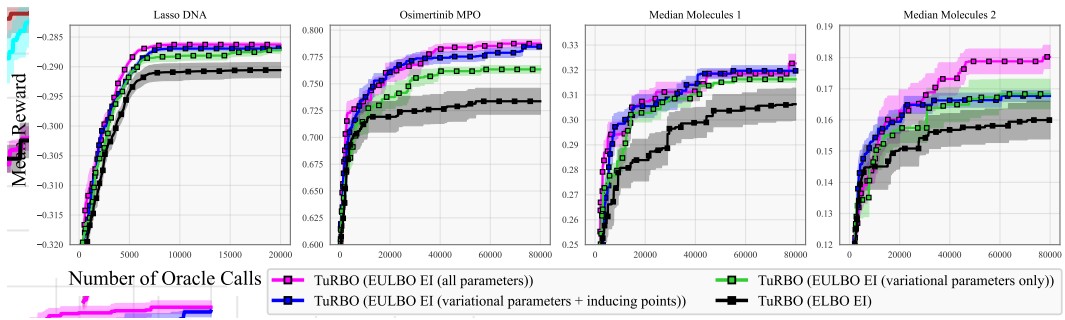

Figure 3: **Ablation study measuring the impact of `EULBO` optimization on various SVGP parameters.** At each BO iteration, we use the standard ELBO objective to optimize the SVGP hyperparameters, variational parameters, and inducing point locations. We then refine some subset of these parameters by further optimizing them with respect to the `EULBO` objective.

**Lasso DNA.** We optimize the 180−dimensional DNA task from the LassoBench library (Šehić et al., 2022) of benchmarks based on weighted LASSO regression (Gasso et al., 2009).

**Molecular design tasks (x4).** We select four challenging tasks from the Guacamol benchmark suite of molecular design tasks (Brown et al., 2019): Osimertinib MPO, Fexofenadine MPO, Median Molecules 1, and Median Molecules 2. We use the SELFIES-VAE introduced by Maus et al. (2022) to enable continuous 256 dimensional optimization.

## 4.2 Optimization Results

In Figure 2, we plot the reward of the best point found by the optimizer after a given number of function evaluations. Error bars show the standard error of the mean over 20 replicate runs. `EULBO` with `TuRBO` outperforms the other baselines with `TuRBO`. Similarly, `EULBO` with standard BO outperforms the other standard BO baselines. One noteworthy observation is that neither acquisition function appears to consistently outperform the other. However, `EULBO`-SVGP almost always dominates ELBO-SVGP and often requires a small fraction of the number of oracle calls to achieve comparable performance. These results suggest that coupling data acquisition with approximate inference/model selection results in significantly more sample-efficient optimization.

## 4.3 Ablation Study

While the results in Fig. 2 demonstrate that `EULBO`-SVGP improves the BO performance it is not immediately clear to what extent joint optimization modifies the posterior approximation beyond what is obtained by standard ELBO optimization. To that end, in Fig. 3 we refine an ELBO-SVGP model with varying degrees of additional `EULBO` optimization. At every BO iteration we begin by obtaining a SVGP model (where the variational parameters, inducing point locations, and GP hyperparameters are all obtained by optimizing the standard ELBO objective). We then refine some subset of parameters (either the inducing points, the variational parameters, the GP hyperparameters, or all of the above) through additional optimization with respect to the `EULBO` objective. Interestingly, we find that tasks respond differently to the varying levels of `EULBO` refinement. In the case of Lasso DNA, there is not much of a difference between `EULBO` refinement on all parameters versus refinement on the variational parameters alone. On the other hand, the performance on Median Molecules 2 is clearly dominated by refinement on all parameters. Nevertheless, we see that `EULBO` is always beneficial, whether applied to all parameters or some subset.

## 5 Related Work

**Scaling Bayesian Optimization to the Large-Budget Regime.** BO has traditionally been confined to the small-budget optimization regime with a few hundred objective evaluations at most. However, recent interest in high-dimensional optimization problems has demonstrated the need to scale BO to large data acquisition budgets. For problems with $\sim 10^3$ data acquisitions, Hernández-Lobato et al. (2017); Snoek et al. (2015); Springenberg et al. (2016) consider Bayesian neural networks (BNN; Neal, 1996), McIntire et al. (2016) use SVGP, and Wang et al. (2018) turn to ensembles of

subsampled GPs. For problems with $\gg 10^3$ acquisitions, SVGP has become the *de facto* approach to alleviate computational complexity (Griffiths and Hernández-Lobato, 2020; Maus et al., 2022, 2023; Stanton et al., 2022; Tripp et al., 2020; Vakili et al., 2021). As in this paper, many works have proposed modifications to SVGP to improve its performance in BO applications. Moss et al. (2023) proposed an inducing point placement based on a heuristic modification of determinantal point processes (Kulesza and Taskar, 2012), which we used for initialization, while Maddox et al. (2021) proposed a method for a fast online update strategy for SVGPs, which we utilize for the KG acquisition strategy.

**Utility-Calibrated Approximate Inference.** The utility-calibrated VI objective was first proposed by Lacoste–Julien et al. (2011), where they used a coordinate ascent algorithm to maximize it. Since then, various extensions have been proposed: Kuśmierczyk et al. (2019) leverage black-box variational inference (Ranganath et al., 2014; Titsias and Lázaro-Gredilla, 2014); Morais and Pillow (2022) use expectation-propagation (EP; Minka, 2001); Abbasnejad et al. (2015) and Rainforth et al. (2020) employ importance sampling; Cobb et al. (2018) and Li and Zhang (2023) derive a specific variant for BNNs; and (Wei et al., 2021) derive a specific variant for GP classification. Closest to our work is the GP-based recommendation model learning algorithm by Abbasnejad et al. (2013), which sparsifies an EP-based GP approximation by maximizing a utility similar to those used in BO.

# 6 Limitations and Discussion

The main limitation of our proposed approach is increased computational cost. While EULBO-SVGP still retains the $O(m^3)$ computational complexity of standard SVGP, our practical implementation requires a warm-start: first fitting SVGP with the ELBO loss and then maximizing the acquisition function before jointly optimizing with the EULBO loss. Furthermore, EULBO optimization currently requires multiple tricks such as clipping and block-coordinate updates. In future work, we aim to develop a better understanding of the EULBO geometry in order to develop developing more stable, efficient, and easy-to-use EULBO optimization schemes. Nevertheless, our results in Section 4 demonstrate that the additional computation of EULBO yields substantial improvements in BO data-efficiency, a desirable trade-off in many applications. Moreover, EULBO-SVGP is modular, and our experiments capture a fraction of its potential use. It can be applied to any decision-theoretic acquisition function, and it is likely compatible with non-standard Bayesian optimization problems such as cost-constrained BO (Snoek et al., 2012), causal BO (Aglietti et al., 2020), and many more.

More importantly, our paper highlights a new avenue for research in BO, where surrogate modeling, approximate inference, and data selection are jointly determined from a unified objective. Extending this idea to GP approximations beyond SVGP and acquisition functions beyond EI/KG may yield further improvements, especially in the increasingly popular high-throughput BO setting.

## Acknowledgments and Disclosure of Funding

The authors thank the anonymous reviewers for suggestions that improved the quality of the work.

N. Maus was supported by the National Science Foundation Graduate Research Fellowship; K. Kim was supported by a gift from AWS AI to Penn Engineering's ASSET Center for Trustworthy AI; G. Pleiss was supported by NSERC and the Canada CIFAR AI Chair program; J. P. Cunningham was supported by the Gatsby Charitable Foundation (GAT3708), the Simons Foundation (542963), the NSF AI Institute for Artificial and Natural Intelligence (ARNI: NSF DBI 2229929), and the Kavli Foundation; J. R. Gardner was supported by NSF awards IIS-2145644 and DBI-2400135.

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

# A  Implementation Details

We will now provide additional details on the implementation. For the implementation, we treat the SVGP parameters, such as the variational parameters $\lambda$, inducing point locations $Z$, and hyperparameters $\theta$, equally. Therefore, for clarity, we will collectively denote them as $w = (\lambda, Z, \theta)$ such that $w \in \mathcal{W} \triangleq \Lambda \times \mathcal{X}^m \times \Theta$, and the resulting SVGP variational approximation as $q_w$. Then, the ELBO and EULBO are equivalently denoted as follows:

$$\mathcal{L}_{\text{ELBO}}(w; \mathcal{D}) \triangleq \mathcal{L}_{\text{ELBO}}(\lambda, Z, \theta; \mathcal{D})$$

$$\mathcal{L}_{\text{EULBO}}(x, w; \mathcal{D}_x, \mathcal{D}_w) \triangleq \mathbb{E}_{f \sim q_w(f)} \log u(x, f; \mathcal{D}_x) + \mathcal{L}_{\text{ELBO}}(w; \mathcal{D}_w).$$

Also, notice that the $\mathcal{L}_{\text{EULBO}}$ separately denote the dataset to be passed to the utility and the ELBO. (Setting $\mathcal{D}_t = \mathcal{D}_w = \mathcal{D}_x$ retrieves the original formulation in Eq. (8).)

**Alternating Updates**  We perform block-coordinate ascent on the EULBO by alternating between maximizing over $x$ as $w$. Using vanilla gradient descent, the $x$-update is equivalent to

$$x \leftarrow x + \gamma_x \nabla_x \mathcal{L}_{\text{EULBO}}(x, w; \mathcal{D}) = x + \gamma_x \nabla_x \mathbb{E}_{f \sim q_w(f)} \log u(x, f; \mathcal{D}),$$

where $\gamma_x$ is the stepsize. On the other hand, for the $w$-update, we subsample the data such that we optimize the ELBO over a minibatch $S \subset \mathcal{D}$ of size $B = |S|$ as

$$w \leftarrow w + \gamma_w \nabla_w \mathcal{L}_{\text{EULBO}}(x, w; S, \mathcal{D}) = w + \gamma_w \nabla_w \left( \mathbb{E}_{f \sim q_w(f)} \log u(x, f; \mathcal{D}) + \mathcal{L}_{\text{ELBO}}(w; S) \right),$$

where $\gamma_w$ is the stepsize. Naturally, the $w$-update is stochastic due to minibatching, while the $x$-update is deterministic. In practice, we leverage the Adam update rule (Kingma and Ba, 2015) instead of simple gradient descent. Together with gradient clipping, this alternating update scheme is much more robust than jointly updating $(x, w)$.

---

**Algorithm 1:** EULBO Maximization Policy

---

**Input:** SVGP parameters $w_0 = (\lambda_0, Z_0, \theta_0)$, Dataset $\mathcal{D}_t$, BO utility function $u$,
**Output:** BO query $x_{t+1}$

1
  ▷ Compute Warm-Start Initializations
2  $w \leftarrow \arg\max_{w \in \mathcal{W}} \mathcal{L}_{\text{ELBO}}(w; \mathcal{D}_t)$ with $w_0$ as initialization.
3  $x \leftarrow \arg\max_{x \in \mathcal{X}} \int u(x, f; \mathcal{D}_t) q_w(f) \, df$
4
  ▷ Maximize EULBO
5 **repeat**
     ▷ Update posterior approximation $q_w$
6    Fetch minibatch $S$ from $\mathcal{D}_t$
7    Compute $g_w \leftarrow \nabla_w \mathcal{L}_{\text{EULBO}}(x, w; S, \mathcal{D}_t)$
8    Clip $g_w$ with threshold $G_{\text{clip}}$
9    $w \leftarrow \text{AdamStep}_{\gamma_w}(w, g_w)$
10
     ▷ Update BO query $x$
11   Compute $g_x \leftarrow \nabla_x \mathcal{L}_{\text{EULBO}}(x, w; S, \mathcal{D}_t)$
12   Clip $g_x$ with threshold $G_{\text{clip}}$
13   $x \leftarrow \text{AdamStep}_{\gamma_x}(x, g_x)$
14   $x \leftarrow \text{proj}_{\mathcal{X}}(x)$
15 **until** *until converged*
16 $x_{t+1} \leftarrow x$
17

---

**Overview of Pseudocode.**  The complete high-level view of the algorithm is presented in Algorithm 1, except for the acquisition-specific details. $\text{AdamStep}_\gamma(x, g)$ applies the Adam stepsize rule (Kingma and Ba, 2015) to the current location $x$ with the gradient estimate $g$ and the stepsize $\gamma$. In practice, Adam is a "stateful" optimizer, which maintains two scalar-valued states for each scalar parameter. For this, we re-initialize the Adam states at the beginning of each BO step.

**Initialization.**    In the initial BO step $t = 0$, we initialize $\boldsymbol{Z}_0$ with the DPP-based inducing point selection strategy of Moss et al. (2023). For the remaining SVGP parameters $\boldsymbol{\lambda}_0$ and $\boldsymbol{\theta}_0$, we used the default initialization of GPyTorch (Gardner et al., 2018). For the remaining BO steps $t > 0$, we use $\boldsymbol{w}$ from the previous BO step as the initialization $\boldsymbol{w}_0$ of the current BO step.

**Warm-Starting.**    Due to the non-convexity and multi-modality of both the ELBO and the acquisition function, it is critical to appropriately initialize the EULBO maximization procedure. As mentioned in Section 3.5, to warm-start the EULBO maximization procedure, we use the conventional 2-step scheme Eq. (5), where we maximize the ELBO and then maximize the acquisition function. For ELBO maximization, we apply Adam (Kingma and Ba, 2015) with the stepsize set as $\gamma_w$ until the convergence criteria (described below) are met. For acquisition function maximization, we invoke the highly optimized `BoTorch.optimize.optimize_acqf` function (Balandat et al., 2020).

**Minibatch Subsampling Strategy.**    As commonly done, we use the reshuffling subsampling strategy where the dataset $\mathcal{D}_t$ is shuffled and partitioned into minibatches of size $B$. The number of minibatches constitutes an "epoch." The dataset is reshuffled/repartitioned after going through a full epoch.

**Convergence Determination.**    For both maximizing the ELBO during warm-starting and maximizing the EULBO, we continue optimization until we stop making progress or exceed $k_{\text{epochs}}$ number of epochs. That is if the ELBO/EULBO function value fails to make progress for $n_{\text{fail}}$ number of steps.

Table 1: Configurations of Hyperparameters used for the Experiments

| Hyperparameter | Value | Description |
|---|---|---|
| $\gamma_{\boldsymbol{x}}$ | 0.001 | ADAM stepsize for the query $\boldsymbol{x}$ |
| $\gamma_{\boldsymbol{w}}$ | 0.01 | ADAM stepsize for the SVGP parameters $\boldsymbol{w}$ |
| $B$ | 32 | Minibatch size |
| $G_{\text{clip}}$ | 2.0 | Gradient clipping threshold |
| $k_{\text{epochs}}$ | 30 | Maximum number of epochs |
| $n_{\text{fail}}$ | 3 | Maximum number of failure to improve |
| $m$ | 100 | Number of inducing points |
| $n_0 = \lvert \mathcal{D}_0 \rvert$ | 100 | Number of observations for initializing BO |
| # quad. | 20 | Number of Gauss-Hermite quadrature points |
| `optimize_acqf: restarts` | 10 | |
| `optimize_acqf: raw_samples` | 256 | |
| `optimize_acqf: batch_size` | 1/20 | Depends on task; see details in Section 4 |

**Hyperparameters.**    The hyperparameters used in our experiments are organized in Table 1. For the full-extent of the implementation details and experimental configuration, please refer to the supplementary code.

# B  Additional Plots

We provide additional results and plots that were omitted from the main text.

## B.1  Additional Results on Molecule Tasks

In Fig. 4, we provide plots on additional results that are similar to those in Fig. 2. On three of the molecule tasks, we use 10,000 random molecules from the GuacaMol dataset as initialization. This is more consistent with what has been done in previous works and achieves better overall optimization performance.

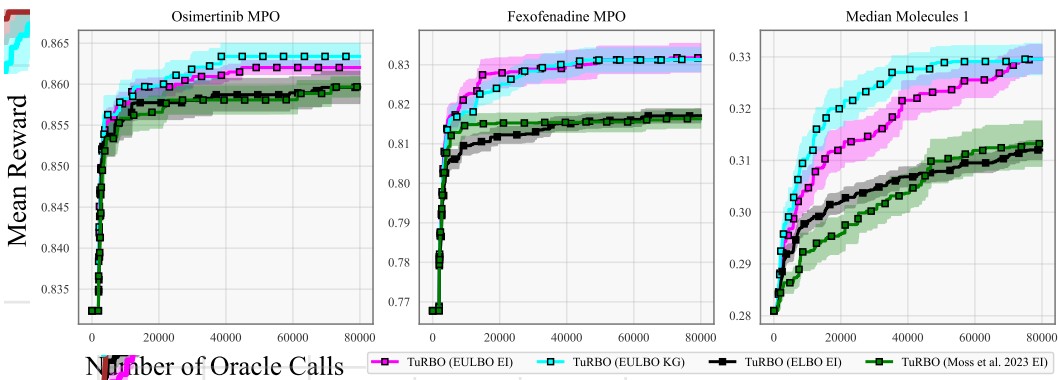

Figure 4: **Additional optimization results on three molecule tasks using 10,000 random molecules from the GuacaMol dataset as initialization**. Each line/shaded region represents the mean/standard error over 20 runs. We count oracle calls starting *after* these initialization evaluations for all methods.

## B.2  Separate Plots for BO and `TuRBO` Results

In this section, we provide additional plots separating out BO and `TuRBO` results to make visualization easier.

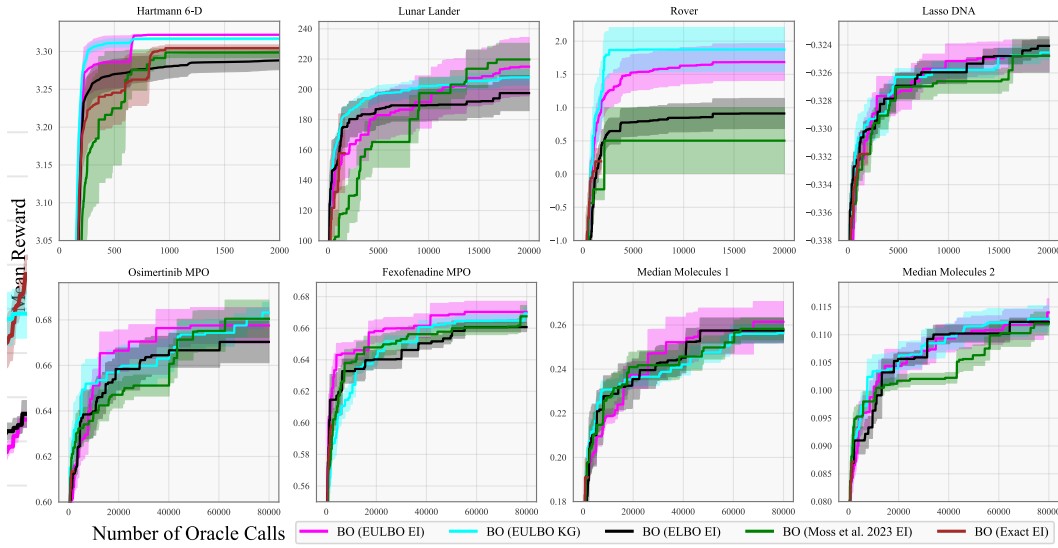

Figure 5: **BO-only optimization results of Fig. 2**. We compare EULBO-SVGP, ELBO-SVGP, ELBO-SVGP with DPP inducing point placement (Moss et al., 2023), and exact GPs. These are a subset of the same results shown in Fig. 2. Each line/shaded region represents the mean/standard error over 20 runs.

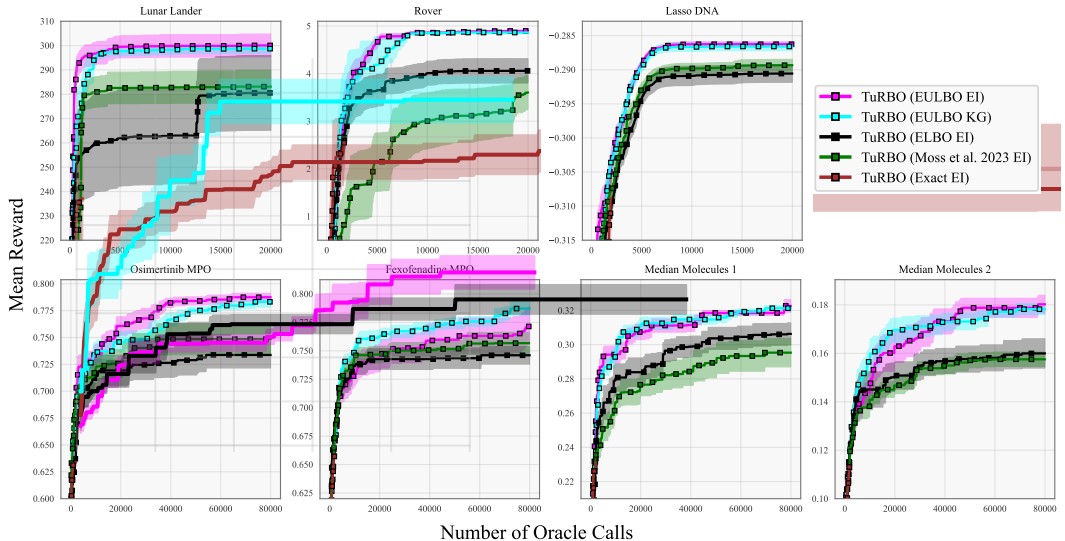

Figure 6: **TuRBO-only optimization results of Fig. 2**. We compare EULBO-SVGP, ELBO-SVGP, ELBO-SVGP with DPP inducing point placement (Moss et al., 2023), and exact GPs. These are a subset of the same results shown in Fig. 2. Each line/shaded region represents the mean/standard error over 20 runs.

### B.3 Effect of Number of Inducing Points

For the results with approximate-GPs in Section 4, we used $m = 100$ inducing points. In Fig. 7, we evaluate the effect of using a larger number of inducing points ($m = 1024$) for EULBO-SVGP and ELBO-SVGP.

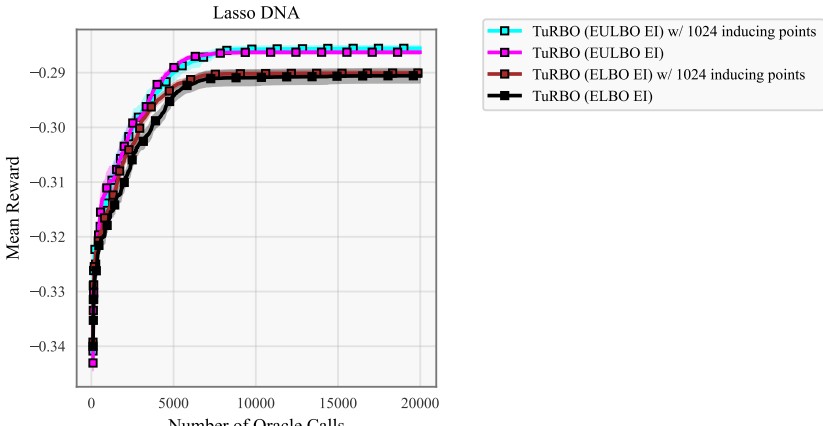

Figure 7: **Ablating the number of inducing points used by EULBO-SVGP and ELBO-SVGP**. As in Fig. 2, we compare running TuRBO with EULBO-SVGP and with ELBO-SVGP using $m = 100$ inducing points used for both methods. We add two additional curves for TuRBO with EULBO-SVGP and TuRBO with ELBO-SVGP using $m = 1024$ inducing points. Each line/shaded region represents the mean/standard error over 20 runs.

Fig. 7 shows that the number of inducing points has limited impact on the overall performance of TuRBO, and EULBO-SVGP outperforms ELBO-SVGP regardless of the number of inducing points used.

## B.4 Effect of GP Objective

The results in Section 4 used a standard SVGP objective. In this section, we evaluate the effect of using an alternative objective: the parametric Gaussian process regressor (PPGPR; Jankowiak et al., 2020) objective. PPGPR differs from the standard SVGP objective in that the variational approximation is optimized to maximize the predictive accuracy instead of matching the posterior.

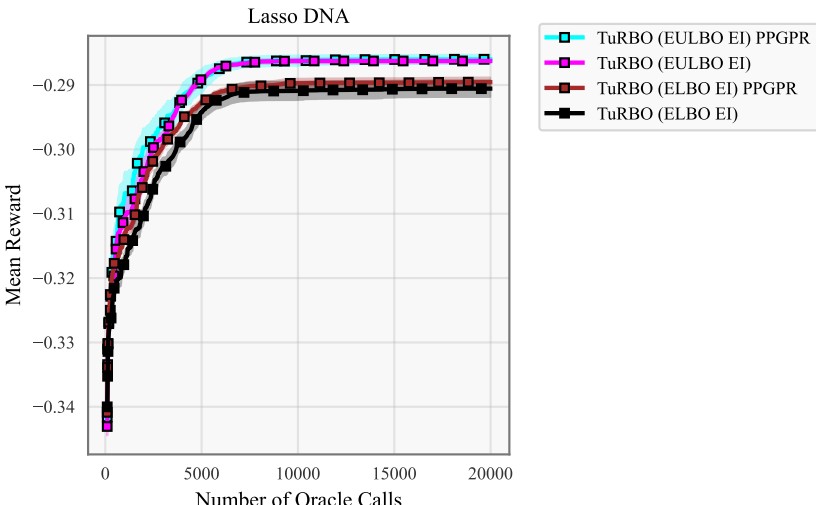

Figure 8: **Effect of using the PPGPR objective instead of the SVGP objective for `EULBO-EI` and `ELBO-EI`**. As in Fig. 2, we compare running `TuRBO` with `EULBO-EI` and with `ELBO-EI` using an SVGP model for both methods. We add two additional curves for `TuRBO` with `EULBO-EI` with a PPGPR model, and `TuRBO` with `ELBO-EI` using a PPGPR model. Each line/shaded region represents the mean/standard error over 20 runs.

We compare the choice of objective (PPGPR vs SVGP) in Fig. 8 and observe that the objective has limited impact on the overall performance of `TuRBO`. In particular, `EULBO-EI` outperforms `ELBO-EI` regardless of the GP objective.

## C    Compute Resources

Table 2: Internal Cluster Setup

| Type | Model and Specifications |
|------|--------------------------|
| System Topology | 20 nodes with 2 sockets each with 24 logical threads (total 48 threads) |
| Processor | 1 Intel Xeon Silver 4310, 2.1 GHz (maximum 3.3 GHz) per socket |
| Cache | 1.1 MiB L1, 30 MiB L2, and 36 MiB L3 |
| Memory | 250 GiB RAM |
| Accelerator | 1 NVIDIA RTX A5000 per node, 2 GHZ, 24GB RAM |

**Type of Compute and Memory.**    All results in the paper required the use of GPU workers (one GPU per run of each method on each task). The majority of runs were executed on an internal cluster, where details are shown in Table 2, where each node was equipped with an NVIDIA RTX A5000 GPU. In addition, we used cloud compute resources for a short period leading up to the subsmission of the paper. We used 40 RTX 4090 GPU workers from `runpod.io`, where each GPU had approximately 24 GB of GPU memory. While we used 24 GB GPUs for our experiments, each run of our experiments only requires approximately 15 GB of GPU memory.

**Execution Time.**    Each optimization run for non-molecule tasks takes approximately one day to finish. Since we run the molecule tasks out to a much larger number of function evaluations than other tasks (80000 total function evaluations for each molecule optimization task), each molecule optimization task run takes approximately 2 days of execution time. With all eight tasks, ten methods run, and 20 runs completed per method, results in Fig. 2 include 1600 total optimization runs (800 for molecule tasks and 800 for non-molecule tasks). Additionally, the two added curves in each plot in Fig. 3 required 160 additional runs (120 for molecule tasks and 40 for non-molecule task). Completing all of the runs needed to produce all of the results in this paper therefore required roughly 2680 total GPU hours.

**Compute Resources Used During Preliminary Investigations.**    In addition to the computational resources required to produce experimental results in the paper discussed above, we spent approximately 500 hours of GPU time on preliminary investigations. This was done on the aforementioned internal cluster shown in Table 2.

## D    Wall-clock Run Times

In Table 3, we provide average wall-clock run times of different methods on the Lasso DNA optimization task.

Table 3: Average wall-clock run times for one full run of TuRBO on the Lasso DNA task. We compare the average wall-clock run time of TuRBO on all TuRBO methods from Figure 2. Note that we do not include the wall clock run time for TuRBO with Exact EI here because we only ran this method out to 2k oracle calls (rather than the full budget of 20k oracle calls).

| Method | Wall-clock Run Time in Minutes |
|--------|-------------------------------|
| EULBO EI | 267.30 ± 2.53 |
| EULBO KG | 296.95 ± 1.31 |
| ELBO EI | 184.40 ± 0.59 |
| Moss et al. 20203 EI | 194.32 ± 0.77 |

