# OpenReview forum: "Approximation-Aware Bayesian Optimization"
_NeurIPS.cc/2024/Conference — NeurIPS 2024 spotlight_

### Official Review · Reviewer_6opG · 2024-06-25

**Soundness:** 4
**Presentation:** 4
**Contribution:** 3
**Rating:** 8
**Confidence:** 3

**Summary:**

The authors present an extension of the sparse Variational Gaussian Process framework (SVGP) that is better suited for Bayesian optimization (BO) tasks. This is achieved by a new optimization criterion (EULBO) which aims to optimize the parameters of SVGPs s.t. the training data is fit well while achieving a better performance in selecting new solution candidates. Two concrete instantiations of EULBO (with Expected Improvement and Knowledge Gradient) are derived. An extensive empirical evaluation is conducted to demonstrate the effectiveness of the proposed approach.

**Strengths:**

- though being dense, the paper is clearly written and straightforward to follow
- the proposed approach is technically solid and mathematically sound
- a relevant problem is addressed by an elegant solution
- both, methodology and experiments are presented well

**Weaknesses:**

## Methodology
- there are some details unclear/missing regarding figure 1:
	- which points were used as inducing points?
	- why is the uncertainty relatively constant at the right side of the plot (left plot)? Depending on the choice of the inducing points, I'd expect the predicted variance (and thus the EI) to increase towards the right edge of the plot
	- what was the exact problem definition leading to Fig. 1? It would be good to see the exact parameterizations for reproducibility as it would help to gauge whether the motivational example is rather cherry-picked or whether the shown issue manifests across different parameterizations
- missing $t$ in Eq. 2? shouldn't it be $\pi(f | \mathcal{D}_t)$?
- in the equation after line 221 it seems that the sum does not depend on $j$. shouldn't it be $\mathbf{x}_j$?

## Experiments
- why were the baselines only applied in conjunction with EI? (see line 259)
- it seems that, depending on the choice of the objective and method, there are quite significant differences in the variance of EULBO. For example, in Fig. 2 "Media Molecules 1", TuRBO + ELBO EI has much higher variance than TuRBO + EULBO EI. Do you account this observation to EULBO?

**Questions:**

see weaknesses

**Limitations:**

the authors discussed the limitations of their approach in a detailed waysssssssssssssssssssssss

---

> ### Author Rebuttal · Authors · 2024-08-07
>
> > Figure 1: which points were used as inducing points?
>
> The inducing points are initialized to a set of points selected uniformly at random in the search space and then they are learned via gradient descent by minimizing the loss (ELBO or EULBO) when the GP model is trained on the data.
>
> > Figure 1: why is the uncertainty relatively constant at the right side of the plot (left plot)? Depending on the choice of the inducing points, I'd expect the predicted variance (and thus the EI) to increase towards the right edge of the plot
>
> In the large interval between the rightmost inducing points, the variance remains fairly constant at the prior variance. At the rightmost inducing point, the variance dips, and then to the right of this rightmost inducing point, the variance increases back up to the prior variance. We think this is fairly expected behavior, and perhaps the dip in variance at the rightmost inducing point was missed due to the relatively large observation noise in this data? We are happy to discuss this with you in more detail.
>
> > Figure 1: what was the exact problem definition leading to Fig. 1? It would be good to see the exact parameterizations for reproducibility as it would help to gauge whether the motivational example is rather cherry-picked or whether the shown issue manifests across different parameterizations
>
> We agree that reproducibility is important for this example problem as well as for other experiments. We plan to include a simple notebook with code to quickly reproduce Figure 1 in our code release which will be available on github once the paper is accepted. We specifically designed the toy dataset to illustrate the point of the paper (large amount of data on the left, very little on the right so that an SVGP with only 4 inducing points would underfit on the right so that the large objective value out right would be missed). Our hope is that this figure is illustrative and aids reader understanding, which we will make more clear in the next version. Of course, the actual experimental results in the paper are actual challenging benchmark problems and not designed for our method. Thank you for the suggestion!
>
> > missing 𝑡 in Eq. 2? shouldn't it be 𝜋(𝑓|𝐷𝑡)?
>
> Thanks for catching this. We will fix this in the next version.
>
> > in the equation after line 221 it seems that the sum does not depend on 𝑗. shouldn't it be 𝑥𝑗?
>
> The maximum (denoted by “max”; not “maximize”!) operation in Line 221 is taken over j.
>
> > why were the baselines only applied in conjunction with EI? (see line 259)
>
> EI is the most commonly used acquisition function used in the BO literature, often because it is cheap to compute and performs comparably with other KG and other acquisition functions. A common motivation in the literature for choosing EI over KG is because KG is significantly more expensive to compute. Running EULBO provides a unique opportunity to make computing KG much more efficient such that it becomes approximately as cheap as EULBO with EI  (see last paragraph of section 3.3). This is not the case for other baselines so running them with KG wouldn’ve been much more expensive. Beyond KG and EI, we note that the EULBO is limited by construction to acquisition functions that admit decision theoretic formulations, i.e. they must be phrasable as posterior expected utilities. This appears to exclude some popular acquisition functions like UCB.
>
> > it seems that, depending on the choice of the objective and method, there are quite significant differences in the variance of EULBO. For example, in Fig. 2 "Media Molecules 1", TuRBO + ELBO EI has much higher variance than TuRBO + EULBO EI. Do you account this observation to EULBO?
>
> This is not something that we have explored specifically, but one plausible explanation is that EULBO consistently converges to a high-performing region of the search space with similar high-scoring molecules, while ELBO converges to a wider variety of lower-scoring regions of the search space, thus resulting in higher variance and lower final objective values. In other words, EULBO may just have lower variance because it consistently performs better on this problem.

---

> > ### Comment · Reviewer_6opG · 2024-08-08
> > **Answer to Rebuttal**
> >
> > Thank you very much for addressing the points I raised!
> >
> > The rebuttal clarified my points.
> >
> > There is only one question left from my side regarding motivation: Could you provide intuition on whether there are certain circumstances in which using ELBO instead of EULBO heavily leads to performance deterioration of BO with SVGPs? In the experiments, one can see that there are problem instances in which EULBO leads to significantly better results than ELBO. Still, in other problem instances, the difference is not that large.
> >
> > Overall, I'm happy to keep my score.

---

> > > ### Author Response · Authors · 2024-08-08
> > > **Regarding your question about motivation**
> > >
> > > This is a good question that we're not sure we have enough data to confidently answer. The benchmark problems we consider in this paper are a mix of problems that some were originally introduced in BO papers using non variational GPs, and others using them. For the ones that have already used variational GPs, there is obviously some survivorship bias where, if SVGPs catastrophically failed on those problems, we wouldn't have seen them. The other problems that originally did not use variational GPs like Rover give us at least some evidence about the performance degradation when switching to SVGPs with ELBO, and it at least does not seem to be catastrophic. We haven't encountered a problem where SVGPs with the ELBO have totally failed compared to known exact GP results, in part because they are often used in situations where evaluation budgets mandate them.

---

> > > > ### Comment · Reviewer_6opG · 2024-08-09
> > > > **Answer to question about motivation**
> > > >
> > > > Thank you for the detailed answer, this clarifies my question.

---

### Official Review · Reviewer_jWXp · 2024-06-27

**Soundness:** 3
**Presentation:** 2
**Contribution:** 4
**Rating:** 7
**Confidence:** 3

**Summary:**

The paper proposes a new method for training sparse Gaussian processes (GPs) for large-budget Bayesian Optimization (BO), designed to facilitate the sequential decision tasks inherent in BO. This is achieved by introducing the “expected utility lower bound (EULBO),” which formulates sparse GP training as a joint optimization problem of BO queries and posterior approximation. The proposed framework is compatible with other significant advancements in the field, such as Expected Improvement (EI) and Knowledge Refinement (KR) acquisition functions, TuRBO, and batch optimization. This compatibility was demonstrated across several BO benchmarks.

**Strengths:**

The method proposed in the paper is novel and original. One of the main strengths is how the newly proposed EULBO integrates with important points in the literature. The authors effectively demonstrate how EULBO can be used in various contexts, such as with EI, KR acquisitions, and batch optimization. This helps distinguish their work from others in the field. The submission is technically sound, with derivations and experiments convincingly supporting the claims of the paper. The work is significant, and I anticipate it will be extended in the future or that some of the ideas presented will be adopted elsewhere.

**Weaknesses:**

While the paper is generally well-written, many derivations were abbreviated and some of the notation was not properly explained, making the paper more difficult to read than necessary. These issues, however, are easily amendable. I provide some suggestions for improvement in the following subsections.

**Questions:**

Introduction:

(line 51): '\emph{minorize-maximize}'

Section 2:

(Fig. 1): It would be beneficial to show the SVGP model fit for EULBO as well.

(line 110): Could you define \lambda, m and S?

(Eq. between 114-115): q_\lambda is initially presented with one argument, then two. If the intention is for q_\lambda to represent all approximate distributions  (in this case, p(f, u) ), please state this explicitly. It is natural to assume that q_\lambda(.) is just the Gaussian pdf (as defined in line 110), which creates confusion.

(line 116): k(.,.) as a function also needs to be explicitly defined.

Section 3:

(line 123): This should probably refer to the equation between lines 114-115, not Equation 3.

(Eq 4): Please define \Lambda.

(Eq 6): Please use double integrals to indicate nested integration. Additionally, I don't understand the emergence of l(D_t |f) and the normalizing constant Z. I was under the impression that l(.|.) is the Gaussian likelihood as defined in line 113. Notice that D_t, as defined in line 92, contains both x and y.

(line 147): 'corresponds to'

(line 165): Please clarify where exactly the requirement for the utility to be strictly positive plays a role or comes from.

(lines 173-175): Would using Gauss-Hermite quadrature for EI make EULBO significantly slower than ELBO?

(line 189): What is y^{+}_t and what its relation to y_t?

Question:

Because you formulate the BO queries and GP approximation as a joint optimization problem without necessarily reducing the dimensionality of the problem, how much slower is this method compared to ‘approximation-ignorant’ BO?

**Limitations:**

The authors are open about the limitations of their work, which they leave for future exploration. Specifically, the increased computational complexity seems to be a primary limitation. It would be beneficial if the authors could quantify this increased complexity, perhaps through reporting empirical runtimes or other relevant metrics.

---

> ### Author Rebuttal · Authors · 2024-08-07
>
> > (line 110): Could you define \lambda, m and S?
>
> $m$ and $S$ are defined as the (learned) mean and covariance of the variational distribution $q(u)$ in our SVGP model. $\lambda$ should have been defined as $\lambda = (m, S)$, a shorthand for “all of the variational parameters,” which we omitted. We will fix this in the next version, and clarify the definition of $m, S$.
>
> > (Eq. between 114-115): q_\lambda is initially presented with one argument, then two. If the intention is for q_\lambda to represent all approximate distributions (in this case, p(f, u) ), please state this explicitly. It is natural to assume that q_\lambda(.) is just the Gaussian pdf.
>
> In Bayesian methods, it is fairly typical to overload the notation of a pdf/measure with its argument. Therefore, $q(u)$ and $q(f, u)$, should be without ambiguity as per common practice. The subscript \lambda is to denote that the distribution denoted as q contains trainable parameters in $\lambda$. We will clarify this.
>
> > (line 116): k(.,.) as a function also needs to be explicitly defined.
>
> Thank you for pointing this out. We will define the kernel function properly in the next version.
>
> > (line 165): Please clarify where exactly the requirement for the utility to be strictly positive plays a role or comes from.
>
> This is due to the log in Eq 6. Thank you for pointing this out, this is pretty subtle and we don’t draw much attention to it. We will clarify this requirement in the next version.
>
> > (lines 173-175): Would using Gauss-Hermite quadrature for EI make EULBO significantly slower than ELBO?
>
> This is a great question that we missed addressing in the original text. Very crucially, the expensive $K_{zz}^{-1}m$ and $K_{zz}^{-1}SK_{zz}^{-1}$ (or $K_{zz}^{-1/2}$ with whitening) solves that dominate both the asymptotic and practical running time of both the ELBO and the EULBO are fixed across the log utility evaluations needed by quadrature (and Monte Carlo in the q-EULBO case). As a result, the additional work of quadrature is pretty negligible. Because Gauss-Hermite quadrature converges extremely quickly in the number of quadrature sites, it only requires on the order of 10 or so of these post-solve evaluations to achieve near machine precision. We will add an explanation of this.
>
> > (Eq 6): Please use double integrals to indicate nested integration. Additionally, I don't understand the emergence of l(D_t |f) and the normalizing constant Z. I was under the impression that l(.|.) is the Gaussian likelihood as defined in line 113. Notice that D_t, as defined in line 92, contains both x and y.
>
> Yes, that is correct. $\pi(f,u|D)$ denotes the “normalized” posterior, which is essentially $\ell(D|f,u) p(f,u) / Z$. Plugging these into the derivation of Eq 6 should clarify things.
>
> > (line 189): What is y^{+}_t and what its relation to y_t?
>
> Thank you for catching this! We forgot to define $y^{+}_t$, which is the highest value of $y_t$ observed so far. We will properly define this in the next version.
>
> > Because you formulate the BO queries and GP approximation as a joint optimization problem without necessarily reducing the dimensionality of the problem, how much slower is this method compared to ‘approximation-ignorant’ BO?
>
> In general, you’re right that the added cost of EULBO is largely due to optimization challenges, rather than concerns like quadrature etc. A single EULBO forward and backward calculation has essentially the same cost as an ELBO calculation; however, as we describe in Section 3.5 and our limitations section, we currently “warm start” EULBO optimization by first optimizing the ELBO, leading to increased computational cost. Here are the wall-clock run times for running TuRBO on the Lasso DNA task using the standard ELBO compared to EULBO with EI and EULBO with KG:
>
> | method | execution time (min) |
> |---|---|
> | ELBO | 184.40 $\pm$ 0.59 |
> | EULBO-EI | 267.30 $\pm$ 2.53 |
> | EULBO-KG | 296.95 $\pm$ 1.31 |
>
> For the camera ready version of the paper, we plan to add a table of wall-clock runtimes comparing EULBO to all baselines.

---

> > ### Comment · Reviewer_jWXp · 2024-08-09
> > **Rebuttal Acknowledgement**
> >
> > I would like to thank authors for their response and addressing my concerns. I am happy to keep the assessment of their work the same.

---

### Official Review · Reviewer_TCW1 · 2024-07-07

**Soundness:** 3
**Presentation:** 3
**Contribution:** 3
**Rating:** 7
**Confidence:** 4

**Summary:**

This paper proposes a modification to sparse variational Gaussian processes (SVGPs) used in Bayesian optimization (BO) to better align the SVGP posterior approximation with the goal of optimizing an acquisition function. The key idea is to jointly optimize the SVGP and the acquisition function using a unified objective called the expected utility lower bound (EULBO). This approach ensures the posterior approximation is well-suited for the downstream decision-making task. The authors derive efficient EULBO objectives for the expected improvement (EI) and knowledge gradient (KG) acquisition functions and demonstrate improved performance over standard SVGPs on several high-dimensional optimization benchmarks.

**Strengths:**

The paper presents a novel approach to aligning SVGP approximations with the goals of BO by jointly optimizing the posterior approximation and acquisition function. This is a creative combination of ideas from variational inference and decision theory.

The proposed method is well-motivated and grounded in sound theoretical principles. The derivations of the EULBO objectives for EI and KG are clear and technically sound. The experiments are comprehensive, covering a range of high-dimensional optimization tasks, and the results convincingly demonstrate the effectiveness of the approach.

The paper is well-written and easy to follow. The authors provide a clear exposition of the problem, the proposed solution, and the experimental setup. The figures and tables effectively communicate the key results.

Scaling BO to high-dimensional spaces is an important problem, and this work represents a significant step towards more efficient and effective BO in such settings. The proposed approach is general and could potentially be applied to other acquisition functions and GP approximations, making it a valuable contribution to the field.

**Weaknesses:**

The paper focuses specifically on SVGPs, but it would be interesting to explore whether the proposed approach can be extended to other sparse GP approximations, even those without a tractable ELBO. This would broaden the applicability of the method and strengthen the contribution.

In Figure 2, the exact GP performs worse than the proposed method and some other baselines in the BO setting. This is counterintuitive, as one would expect the exact GP to be the gold standard when computationally feasible. The authors should provide more discussion and insights into this unexpected result.

The paper presents several variations of the proposed method (e.g., EI-EULBO, KG-EULBO, batch versions), but it lacks a clear recommendation on which variant to use in practice. A more in-depth discussion of the trade-offs between these variants and guidance on when to use each one would enhance the practical value of the work.

**Questions:**

Can the proposed EULBO approach be extended to other sparse GP approximations beyond SVGPs, even if they do not have a tractable ELBO? If so, what challenges would need to be addressed, and how might the optimization problem be formulated?

Why does the exact GP perform worse than the proposed method and some other baselines in the BO experiments (Figure 2)? Is this due to the limitations of the exact GP in high dimensions, or are there other factors at play?

What are the key factors to consider when choosing between the different variants of the proposed method (e.g., EI-EULBO, KG-EULBO, batch versions)? In what scenarios would one variant be preferred over another?

**Limitations:**

The authors have addressed the limitations of their work in Section 6, acknowledging the increased computational cost of the EULBO approach compared to standard SVGPs. They also mention the need for multiple optimization tricks and the potential instability of the EULBO optimization problem. These are important limitations that users should be aware of when considering this method.

However, the authors do not discuss any potential negative societal impacts of their work. While the proposed method is primarily methodological and does not pose immediate societal risks, it would be valuable for the authors to briefly comment on any broader implications of improved BO in high-dimensional spaces, such as the potential for misuse or unintended consequences in certain application domains.

---

> ### Author Rebuttal · Authors · 2024-08-07
>
> > The paper focuses specifically on SVGPs, but it would be interesting to explore whether the proposed approach can be extended to other sparse GP approximations, even those without a tractable ELBO. This would broaden the applicability of the method and strengthen the contribution.
>
> > Can the proposed EULBO approach be extended to other sparse GP approximations beyond SVGPs, even if they do not have a tractable ELBO? If so, what challenges would need to be addressed, and how might the optimization problem be formulated?
>
> Our focus on SVGP was motivated by the fact that SVGPs are the most widely used sparse GP approximation in the high-throughput BO literature, but your question is a good one. SVGPs have been extended and improved several times, and it’s a natural question whether the EULBO can be adapted to all of these settings. Our method should be readily applicable when an ELBO exists mathematically but can only be approximated through Monte Carlo or quadrature. For sparse approximations where the objective is not a canonical ELBO, it is possible that a similar variational utility lower-bound could be derived, but it may look very different from this paper. We conjecture that some recent SVGP-like models like ODSVGP admit relatively straightforward EULBO adaptations. Other examples, like Vecchia approximations, would require pretty orthogonal machinery to what we introduce and would probably be interesting in their own right. We are happy to provide more detailed comments on specific sparse GP approximations if the reviewer has one in mind, and will add a discussion to the camera ready version.
>
> > In Figure 2, the exact GP performs worse than the proposed method and some other baselines in the BO setting. This is counterintuitive, as one would expect the exact GP to be the gold standard when computationally feasible. The authors should provide more discussion and insights into this unexpected result.
> > Why does the exact GP perform worse than the proposed method and some other baselines in the BO experiments (Figure 2)? Is this due to the limitations of the exact GP in high dimensions, or are there other factors at play?
>
> While we agree that this is counter-intuitive, we point to the fact that the true posterior of an SVGP is not an exact GP. Strictly speaking, SVGPs not only provide a variational approximation, but also modify the GP model to have an additional layer of latent variables. Therefore, there is no reason to expect that the performance of SVGPs would be the same as exact GPs, even if one has access to their true posterior. Additionally, and perhaps most interestingly, Exact GPs arguably have their own “mismatches” between learning and acquisition like the ones pointed out in and addressed by this paper: hyperparameter learning in exact GPs is not done in a utility aware fashion, for example. It seems plausible that, for some problems, utility aware hyperparameter learning might outweigh the loss incurred by using a sparse model. We do note that whether our method is better or worse than exact GPs is unlikely to be consistent across different tasks, and we suspect exact GPs are still quite competitive in settings where they can be used. (Also note the response to Reviewer veTg on a similar question.)
>
> > The paper presents several variations of the proposed method (e.g., EI-EULBO, KG-EULBO, batch versions), but it lacks a clear recommendation on which variant to use in practice. A more in-depth discussion of the trade-offs between these variants and guidance on when to use each one would enhance the practical value of the work.
> > What are the key factors to consider when choosing between the different variants of the proposed method (e.g., EI-EULBO, KG-EULBO, batch versions)? In what scenarios would one variant be preferred over another?
>
> The best choice for which acquisition function to use, whether or not to use batch BO, what batch size to use, etc., often vary widely across problems and it’s hard for us to give guidance beyond what exists in the literature: KG is often less myopic than EI, which can be better on some problems and worse on others. Definitively comparing acquisition strategies probably deserves its own investigation. One thing we do note, however, is that the use of the EULBO switches the balance in the computational-performance trade-off of acquisition functions. For instance, the cost of computing the EULBO amortizes the cost of computing KG, making it a more sensible choice in some settings than it was before (see section 3.3 last paragraph). But again, whether one should use EI or KG given this tipping of the scale will highly depend on the problem and this paper probably doesn’t help definitively answer that question.

---

> > ### Comment · Reviewer_TCW1 · 2024-08-11
> >
> > Thank you for your response, which resolves my concerns and raises some very interesting and insightful discussion on the svgp.

---

### Official Review · Reviewer_veTg · 2024-07-09

**Soundness:** 4
**Presentation:** 4
**Contribution:** 3
**Rating:** 7
**Confidence:** 4

**Summary:**

The paper proposes a new approach for scaling Bayesian Optimisation to large datasets. Contrary to previous approaches, which fitted a sparse GP and optimised acquisition function independently, the paper proposes a method, which jointly optimises the variational parameters of sparse GP and searches for the next point to query. This is done by adding a utility term to ELBO. Authors discuss how this objective can be connected to generalised Bayesian Inference. Authors evaluate the proposed method on a number of benchmark with high number of datapoints and show it outperforms other baselines.

**Strengths:**

- The paper touches on a very important subject of scaling Bayesian Optimisation to large datasets. I find the proposed solution to be very elegant. I also think the connection to generalised Bayesian Inference is very interesting.

- The method seems to be delivering a significant improvement in sample efficiency over baselines. I believe the choice of benchmarks is sufficiently diverse and the number of baselines compared against is sufficient.

- The paper is well-written and I really like that authors share the details of the tricks they used for improved optimisation of EULBO. I also like the fact that authors are very open about admitting the limitations of the proposed method.

**Weaknesses:**

- As mentioned by the authors, optimisation of the EULBO objective is a bit cumbersome and seems this process is also slower than the optimisation of standard ELBO. However, it might be worth to sacrifice a bit of computational efficiency for an increase in the sample complexity.

- While authors cite previous work, which proved that the selected action satisfies convergence guarantee, it does not directly prove anything about the performance of optimisation process choosing action in such a way (e.g. regret bound or convergence to optimum). The paper could be made much stronger if authors managed to connect those notions in some way. However, I also do not think it is critical, as the main contribution of the paper is empirical (although it would be greatly appreciated).

**Questions:**

- Can the authors provide the running time of the proposed method (and baselines) ? It is ok if the proposed method is slower, but it would be good to quantify exactly how much slower it is.

- Do authors have any expectations on how would the method perform in comparison to an exact GP on a smaller dataset? While I understand that the method is particularly designed for large datasets, it would be interesting to see how the utility VI-based acquisition strategy compares to standard acquisition functions like EI. It would be nice to have at least such experiment in the camera-ready version to judge whether the improvement delivered by the method comes from better acquisition strategy or better modelling

**Limitations:**

Authors admit that the increased difficulty of optimising EULBO is a limitation of their work.

---

> ### Author Rebuttal · Authors · 2024-08-07
>
> > While authors cite previous work, which proved that the selected action satisfies convergence guarantee, it does not directly prove anything about the performance of optimisation process choosing action in such a way (e.g. regret bound or convergence to optimum). The paper could be made much stronger if authors managed to connect those notions in some way. However, I also do not think it is critical, as the main contribution of the paper is empirical (although it would be greatly appreciated).
>
> From the best of our knowledge, non-asymptotic convergence proofs under approximate inference, both in Bayesian optimization and bandits are rare. Furthermore, most existing works assume that one can explicitly control the fidelity of the approximation or that the true posterior is within the variational family. The benefits of utility-calibrated variational inference, on the other hand, only exists when the approximation is imperfect. Therefore, we are currently unsure how to analyze the theoretical properties of utility-calibrated Bayesian inference. However, we believe this would be a very interesting avenue for future research and will definitely explore this direction. We also point out that some asymptotic analyses of consistency exist as mentioned in Line 151-152.
>
> > Can the authors provide the running time of the proposed method (and baselines) ? It is ok if the proposed method is slower, but it would be good to quantify exactly how much slower it is.
>
> Here are the wall-clock run times for running TuRBO on the Lasso DNA task from the paper using the standard ELBO compared to using EULBO with EI and using EULBO with KG:
>
> | method | execution time (min) |
> |---|---|
> | ELBO | 184.40 $\pm$ 0.59 |
> | EULBO-EI | 267.30 $\pm$ 2.53 |
> | EULBO-KG | 296.95 $\pm$ 1.31 |
>
> We agree that providing a full table of average wall-clock runtimes for EULBO and all baselines would be useful, we will plan to add this table to the appendix for the camera ready version. Thank you for the suggestion!
>
> > Do authors have any expectations on how would the method perform in comparison to an exact GP on a smaller dataset? While I understand that the method is particularly designed for large datasets, it would be interesting to see how the utility VI-based acquisition strategy compares to standard acquisition functions like EI. It would be nice to have at least such experiment in the camera-ready version to judge whether the improvement delivered by the method comes from better acquisition strategy or better modelling
>
> We point out to the top left pane in Figure 2, where we present results on the classic Hartmann 6 function. Somewhat surprisingly, our proposed approach does better than exact GPs in this one instance. Because SVGPs are not only a variational approximation, but a distinct model from exact GPs due to the use of inducing points, there is no reason to expect that exact GPs are a limiting case for the BO performance of SVGPs. Indeed, Exact GPs arguably have their own “mismatches” between learning and acquisition like the ones pointed out in this paper: hyperparameter learning in exact GPs is not done in a utility aware fashion, for example, while it is with EULBO. In general, we still expect exact GPs to be quite performant in settings where it can be applied. (Also note the response to Reviewer TCW1 on a similar question.)

---

> > ### Comment · Reviewer_veTg · 2024-08-11
> >
> > Thank you very much for responding to my review. I am happy with the response and I believe the paper is worthy of publication.

---

### Decision · Program_Chairs · 2024-09-25

**Decision:**

Accept (spotlight)

**Comment:**

This work scales up Bayesian Optimisation to large datasets. The paper is clear and claims supported. All reviewers were enthusiastic about this work and all concerns raised were adequately addressed during the rebuttal. This is a solid piece of work that is relevant to the community.